# Development of an enhanced analytical method utilizing pepper matrix as an analyte protectant for sensitive GC–MS/MS detection of dimethipin in animal-based food products

**Jae-Han Shim**[1]***, Md. Musfiqur Rahman**[1]**, Tuba Esatbeyoglu**[2]***, Fatih Oz**[3]**, A. M. Abd El-Aty**[4,5]*

**1** Natural Products Chemistry Laboratory, College of Agriculture and Life Sciences, Chonnam National University, Gwangju, Republic of Korea, **2** Department of Food Development and Food Quality, Institute of Food Science and Human Nutrition, Gottfried Wilhelm Leibniz University Hannover, Hannover, Germany, **3** Department of Food Engineering, Faculty of Agriculture, Ataturk University, Erzurum, Turkey, **4** Department of Pharmacology, Faculty of Veterinary Medicine, Cairo University, Giza, Egypt, **5** Department of Medical Pharmacology, Medical Faculty, Ataturk University, Erzurum, Turkey

\* jhshim@jnu.ac.kr (JHS); esatbeyoglu@lw.uni-hannover.de (TE); abdelaty44@hotmail.com (AMAE)

**Data Availability Statement:** All relevant data are within the paper.

## Abstract

Herein, an analytical method using gas chromatography-tandem mass spectrometry (GC–MS/MS) was devised to detect the presence of the troublesome pesticide dimethipin in various animal-based food products, including chicken, pork, beef, eggs, and milk. The injection port was primed with a matrix derived from pepper leaves that acts as an analyte protectant (AP) to safeguard the target compound from thermal degradation during gas chromatography. The presence of AP resulted in a remarkable limit of quantification of 0.005 mg/kg for dimethipin in five matrices. Three different versions (original, EN, and AOAC) of the QuEChERS (Quick, Easy, Cheap, Effective, Rugged, and Safe) method were compared for dimethipin extraction, with a double-layer solid-phase extraction (SPE) cartridge utilized for matrix purification. A seven-point external calibration curve was established for dimethipin in the five matrices, demonstrating excellent linearity with determination coefficients ($R^2$) $\geq$ 0.998. The developed quantitative method was validated by fortifying each matrix with three different concentrations of standard dimethipin, and the average recovery fell within the acceptable range outlined in the CODEX guidelines (ranging from 88.8% to 110.0%), with a relative standard deviation (RSD) of $\leq$ 11.97%. This method effectively addresses the challenge of analyzing dimethipin and can therefore be used as a routine monitoring tool for dimethipin across various matrices.

## Introduction

Dimethipin, also known as 2,3-dihydro-5,6-dimethyl-1,4-dithiin 1,1,4,4-tetraoxide, serves as a plant growth regulator and defoliant used in the agricultural industry for oilseeds, potatoes, and tomatoes [1]. Its primary purpose lies in aiding harvesting procedures by functioning as

**Funding:** The authors received no specific funding for this work.

**Competing interests:** The authors have declared that no competing interests exist.

a desiccant. The mechanism of action involves inducing water loss through stress on the plant's stomatal system, resulting in leaf turgor reduction, desiccation, and leaf abscission. Dimethipin has been officially registered in the United States since 1982 for diverse purposes, encompassing cotton growth regulation, defoliation, and postemergent herbicide applications. Over the years, the Americas have consistently ranked as the leading consumer of pesticides globally, outpacing Asia, Europe, Africa, and Oceania since the mid-1990s. Consequently, the majority of reports tend to focus on states within the Americas rather than other regions [2]. It is classified as having moderate acute toxicity (Category II) when ingested or inhaled [3]. Furthermore, it is considered a class C (possible) human carcinogen, according to the United States Environmental Protection Agency [4]. Due to its relatively high water solubility and low Koc > 1000, dimethipin has notable leaching and groundwater contamination potential. It must be noted that the use of defoliants typically destroys the plant canopy, thereby increasing the likelihood of surface runoff and leaching during subsequent precipitation events [1].

Dimethipin can potentially reach animal-based food products through various pathways. One route is through direct application, where dimethipin is applied to crops that are subsequently used as animal feed. When animals consume these treated crops, residues of dimethipin can be transferred to their tissues [4]. Another pathway is through residue transfer in livestock farming. Suppose dimethipin is used as a pesticide spray or medication for animals. In that case, it can be absorbed by the animals and accumulate in their tissues, including muscles, fat, or organs, ultimately resulting in animal-based food products [2]. Environmental exposure can also contribute to the presence of dimethipin in animal-based food products. Dimethipin can enter the environment through agricultural runoff or the breakdown of treated crops. Animals grazing on fields where dimethipin has been used or drinking water contaminated with the chemical can take up dimethipin, leading to its presence in their bodies and subsequent occurrence in animal-derived food products [5]. Furthermore, feed contamination can be a potential pathway for dimethipin to reach animal-based food products. If animal feed is produced using crops treated with dimethipin, residues of the chemical can be present in the feed itself. When animals consume this contaminated feed, dimethipin can be transferred to their tissues and ultimately found in animal-based food products [6]. The presence of dimethipin residues in animal-based food products has raised concerns due to potential health risks associated with its consumption. On the other hand, determining dimethipin residues in animal-based food products is challenging due to the low levels of residues, complex sample matrices, chemical properties of dimethipin, matrix effects, and regulatory considerations. Therefore, overcoming these challenges requires the development of a sensitive, accurate, and reliable method, which is crucial for food safety, risk assessment, consumer confidence, and quality control. This ensures that dimethipin residues are monitored and controlled within acceptable limits, safeguarding public health and maintaining the integrity of the food supply chain.

Numerous approaches exist in the literature for examining pesticide residues in plants [7, 8] and environmental samples [9, 10]. However, there is limited documentation regarding methodologies designed explicitly for analyzing residue levels in animal-derived food. When dealing with fatty matrices, such as animal-based food, it is essential to develop sample preparation methods that focus on removing lipids and extractives [11, 12]. Consequently, analytical techniques for pesticides in animal-derived food usually incorporate a purification step, such as liquid–liquid extraction (LLE) [13–16], gel permeation chromatography [17], and/or solid-phase extraction (SPE) [18, 19]. However, most of these methods require substantial amounts of solvents and are intricate, labor intensive, and expensive [20]. Additionally, prolonged sample preparation procedures may lead to the degradation of some

pesticides. The streamlined and efficient "QuEChERS" method, known for its quick, easy, cheap, effective, rugged, and safe characteristics, has been widely utilized for extracting pesticides from fruits and vegetables [21]. This method has also been adapted and applied to determine pesticide residues in complex fatty matrices, including fish and fish food [22, 23], eggs and meat [23, 24], bovine milk [25], avocado [26], and peanut oil [27]. On the other hand, continuous advancements are being made in the realm of analytical separation and detection, focusing on the quest for faster methodologies that incorporate liquid chromatography (LC) or gas chromatography (GC) coupled with mass spectrometry (MS) [28, 29]. Narrow-bore capillary columns are increasingly employed in GC and GC–MS, offering notable improvements in both laboratory efficiency and precision while maintaining chromatographic resolution comparable to that of conventional capillary GC [30, 31]. Gas chromatography combined with tandem mass spectrometry (GC–MS/MS) has emerged as a powerful technique for pesticide residue analysis, offering high sensitivity and selectivity. Nonetheless, the accurate detection of dimethipin in complex matrices such as animal-based food products remains challenging due to matrix interferences and analyte degradation during sample preparation and analysis.

A subsequent endeavor by Anastassiades et al. aimed to refine earlier efforts in identifying masking agents capable of shielding active sites within the GC system, thereby yielding substantial response enhancement for pesticides. These specific compounds were coined "analyte protectants" (APs). The utilization of APs offers several key advantages, including simplified preparation of calibration standards and improved accuracy in analysis. The concept of APs has been successfully implemented in the development of the QuEChERS method [21] for the analysis of pesticide residues using GC–MS [32], including direct sample introduction techniques [33].

To the best of the authors' knowledge, only two articles have examined dimethipin in conjunction with other pesticides, utilizing techniques such as GC–NPD, GC–MS, and HPLC–diode array detection to analyze its presence in water [1]. Additionally, GC–MS/MS was employed for residue determination in agricultural products [34]. However, no articles were found that specifically investigated the residue levels of dimethipin in animal-based food products. Therefore, the objective of this study was to modify the QuEChERS method with priming and APs for the analysis of dimethipin in five food products of animal origin, namely, chicken, pork, beef, egg, and milk, utilizing GC–MS/MS. Adding an analyte protectant during sample preparation protects against analyte degradation and enhances its stability throughout the analytical process. This protective effect helps to mitigate the loss or transformation of dimethipin during sample extraction, derivatization, and GC–MS/MS analysis, resulting in improved detection limits and reliable quantification.

## Material and methods

### Chemicals and reagents

The dimethipin standard with a purity of 99% was procured from Dr. Ehrenstorfer (Augsburg, Germany). High-performance liquid chromatography (HPLC)-grade acetonitrile was obtained from Burdick and Jackson (SK Chemical, Ulsan, Republic of Korea). Formic acid was sourced from Junsei Chemical Co., Ltd. (Tokyo, Japan). The QuEChERS European (EN) kit, consisting of 4 g $MgSO_4$, 1 g NaCl, 1 g $Na_3Cit.2H_2O$, and 0.5 g $Na_2Cit.5H_2O$, as well as the Original kit (4 g $MgSO_4$ and 1 g NaCl) and AOAC QuEChERS kit (6 g $MgSO_4$ and 1.5 g NaOAc), were obtained from Agilent Technologies, Inc. (Santa Clara, CA, USA). A double-layer carbon/PSA cartridge (500/500 mg) was purchased from Supelco (Bellefonte, PA, USA).

## Preparation of stock and calibration solutions

A 100 ppm stock solution of dimethipin was prepared by dissolving 10.10 mg of the standard in 100 mL of acetonitrile. Intermediate working standard solutions of 10 ppm were then prepared by diluting the stock solution with the same solvent. To create a solvent calibration curve, the working standard was further diluted in the following order: 0.005, 0.01, 0.015, 0.02, 0.03, 0.05, and 0.06 ppm. The standard stock solution was stored at -24˚C for one month, while the working and calibration solutions were kept at 4˚C until analysis.

## Sample preparation

A total of 5 g of freeze-dried (chicken, pork, and beef) or fresh (egg and milk) samples was placed in a 50 mL Teflon centrifuge tube and spiked with 1 mL of 0.1 mg/kg (1 × limit of quantification [LOQ] level), 0.2 mg/kg (2 × LOQ level), or 1.0 mg/kg (10 × LOQ level). For homogenization (excluding egg and milk), 10 mL of acetonitrile (ACN) was added, and the mixture was homogenized using a tissue homogenizer for 1 min. The QuEChERS EN kit (consisting of 4 g $MgSO_4$, 1 g NaCl, 1 g $Na_3Cit.2H_2O$, and 0.5 g $Na_2Cit.5H_2O$) was then added to the tube. Manual shaking for 1 min and vortex mixing for 1 min were performed, followed by centrifugation at 4,000 rpm for 5 min. From the resulting supernatant, 3.0 mL was loaded onto a double-layer (carbon/PSA, 500/500 mg) cartridge that had been previously conditioned with 6 mL of ACN: toluene (3:1, *v:v*). The target analyte was eluted with 6 mL of ACN:toluene (3:1, *v:v*). The loaded and eluted solutions were combined and concentrated under reduced pressure in a 40˚C water bath. Subsequently, the concentrated solution was redissolved in 1 mL of acetonitrile containing analyte protectant (AP), resulting in a final concentration factor of 0.93 for the sample. The concentration of AP was 30% (0.25 g/mL) for subsequent GC–MS/MS analysis.

The presence of the pepper leaf matrix serves to deactivate active sites within the GC system, ensuring that dimethipin can travel from the injector to the detector through the capillary column without any undesired interaction [35, 36]. Consequently, the pepper leaf matrix was carefully optimized to enhance the response of dimethipin during GC-mass selective detection. This optimization was carried out following the preparation of the leaf matrix using our previously established method [35].

## GC–MS/MS analysis

For the instrumental analysis of dimethipin, an Agilent Intuvo 9000 GC system (Santa Clara, CA, USA) coupled with an Agilent 7000 D GC/MS TQ detector was utilized. The compound was separated using an HP-5 MS capillary column (30 m × 0.25 mm I.D. × 0.25 μm; Intuvo, Santa Clara, CA, USA) in gradient mode. The column oven temperature was initially set at 80˚C for 2 min and then gradually increased to 200˚C at a rate of 20˚C/min, where it was held for 1 min. Subsequently, the temperature was further raised to 300˚C at a rate of 25˚C/min and maintained for 2 min. The liner that was used in the injector port was an Ultra inert liner, splitless single taper liner with glass wool (900 μL, 4 mm id), whereas the septum was an Inlet Septa: Agilent, Bleed/Temp optimized (BTO) nonstick 11 mm, 50/pk. Both the liner and septum are changed twice a month, while the column is replaced monthly (1st cutting). As a carrier gas, helium was employed at a flow rate of 1.5 mL/min. The injection port temperature was set at 270˚C, and a volume of 1 μL was injected. The detector temperature was maintained at 280˚C, operating in electron spray ionization mode. The multiresidue method (MRM) transition utilized two precursor ions, *m/z* 210 and 118. The collision energy for generating the first product ion, 210→76, was set to 7, while the collision energy for the second product ion, 118→58, remained the same. Data acquisition was performed using Mass Hunter software. Ensuring the quality of injections in GC–MS/MS in the absence of a deuterated compound

involves implementing various strategies. First, well-defined, symmetrical peaks with good resolution are indicative of a high-quality injection. Additionally, a stable baseline is crucial, suggesting proper injection and chromatographic conditions. Consistent retention times across injections contribute to the overall reproducibility. Performing blank runs by injecting solvent-only blanks helps identify any potential contamination or interference, with the baseline remaining stable. Periodic injection of standard solutions verifies the instrument's sensitivity and linearity. Matrix effects are assessed by analyzing matrix-matched standards to understand the sample matrix's impact on the analytical signal. Regular injection of calibration standards ensures that the instrument response aligns with the calibration curve. Last, peak identity is confirmed by comparing mass spectra and retention times with reference standards or library spectra.

## Method validation

The validation and optimization of the method followed the guidelines outlined in EU document no. SANTE/11945/2015 (SANTE Guidance Document on Analytical Quality Control and Method Validation Procedures for Pesticide Residue Analysis in Food and Feed, 2015 [37]. The parameters assessed included linearity, matrix effects, limits of detection and quantification, specificity, accuracy, and precision. Linearity was evaluated by calculating the determination coefficient ($R^2$) using a seven-point matrix-matched calibration curve (0.005, 0.01, 0.015, 0.02, 0.03, 0.05, and 0.06 mg/kg). The limit of detection (LOD) was determined as the lowest detectable signal of the analyte that was three times higher than the background noise, following the method described by Ribani et al. [38]. The limit of quantification (LOQ) was defined as the minimum concentration of the analyte that could be accurately and precisely quantified. The specificity of the method was assessed by injecting control samples and observing any interference. Specificity was considered satisfactory if there was either no interference or interference at a level below 30% of the LOQ at the retention times of the target compound. The accuracy determination involved assessing recovery, estimated by fortifying three blank samples with standard solutions at three distinct concentrations (1×, 2×, and 10× LOQ). To measure recoveries, the peak areas of the spiked samples were compared to those of external standards in the matrix. The precision (repeatability, RSD) was calculated to evaluate variations in repeated analyses under the same experimental conditions. On the other hand, the matrix effect (ME) was assessed by comparing the slope of the calibration curve for standard solutions with that of matrix-matched standard solutions. Positive values indicate matrix enhancement, while negative values suggest matrix suppression. Quality controls (QCs) can be implemented through calibration standards, including blanks in each batch to monitor and correct for any contamination or background signals, and matrix-matched standards.

## Results and discussion

### Optimization of different elution volumes

In this study, we investigated the impact of different elution volumes (6 mL, 8 mL, 10 mL, and 12 mL) of ACN:toluene (3:1, *v:v*) on the recovery of dimethipin, both in a solvent and in a matrix (egg) sample. The recovery percentages varied depending on the elution volume used. For the dimethipin standard in the solvent, the recovery ranged from 99–100%, 99–103%, 97–99%, and 97–98% for the respective elution volumes. In the case of dimethipin in the matrix (egg), the recovery percentages ranged from 80–82%, 76–83%, and 78–84% and were unavailable for the different elution volumes. Based on the data obtained, an elution volume of 6 mL yielded a satisfactory recovery for dimethipin in both the solvent and matrix (egg) samples.

These findings highlight the importance of optimizing elution conditions to ensure accurate analytical measurements, as the elution volume influences the recovery percentages.

## Comparison of QuEChERS methods for the recovery of dimethipin

To ensure accurate recovery of the target analyte, selecting the appropriate QuEChERS method is crucial. In this study, we compared the recovery percentages of dimethipin using three different QuEChERS versions: Original, AOAC (Association of Official Analytical Chemists), and EN (European QuEChERS). The data obtained indicate variations in the recovery percentages of dimethipin among these three QuEChERS versions. The original QuEChERS method yielded recovery percentages ranging from 87% to 92%, while the AOAC QuEChERS method showed slightly lower recovery percentages ranging from 82% to 90%. On the other hand, the EN QuEChERS method resulted in the highest recovery percentages, ranging from 88% to 89%. Based on the results, the EN QuEChERS method was selected as the optimal approach for extracting dimethipin from animal-based food products. This method consistently demonstrated the highest recovery percentages among the tested versions, suggesting its effectiveness in accurately extracting dimethipin from food matrices.

## Priming and analyte protectant

For analysis of dimethipin in different matrices, two precursor ions (210 and 118 *m/z*) were chosen for the multiresidue method (MRM) transition. These precursor ions were selected in preparation for subsequent analysis of dimethipin across various matrices using GC–MS/MS. Fig 1 provides clear and concise information about the experimental setup and the observed effects of pepper leaves on the dimethipin signal by GC–MS/MS. This indicates that at a concentration of 0.01 ppm, both with and without the pepper leaf matrix, the dimethipin signal was measured. Additionally, at a higher concentration of 0.06 ppm with the pepper leaf matrix, signal suppression of dimethipin was observed after 3 h of continuous batch running.

As implied in Fig 2, adding an analyte protectant (AP) after priming with a pepper leaf matrix can significantly enhance the signal of dimethipin in GC–MS/MS analysis. Furthermore, the signal enhancement level increases with higher AP concentrations, with the highest enhancement observed at a 30% AP concentration. These findings demonstrate the importance of optimizing the use of an analyte protectant to improve the detection and quantification of dimethipin in the presence of the pepper leaf matrix. The matrix of pepper leaves potentially contains a higher number of compounds than pepper itself, which can render the GC system's active site inactive. This active site is responsible for the decomposition of analytes [35]. In this context, kresoxim-methyl and its two thermolabile metabolites, BF 490–2 and BF 490–9, were assessed in pear samples using a pepper leaf matrix as a protective measure to preserve these metabolites during GC analysis. Incorporating the pepper leaf matrix into the pear matrix ensured the integrity of the compounds, resulting in a precise and sensitive GC outcome for thermolabile kresoxim-methyl metabolites. The limits of detection (LOD) and quantification (LOQ) were determined to be 0.006 and 0.02 mg/kg for kresoxim-methyl and 0.02 and 0.065 mg/kg for the metabolites, respectively [39].

From the author's experience, pepper leaf as an AP over alternatives, such as 3-ethoxy-1,2-propanediol or D-sorbitol, may offer the following advantages: 1- Pepper leaf is a natural, plant-derived material, which might make it more compatible with certain analytes or matrices compared to synthetic compounds such as 3-ethoxy-1,2-propanediol. 2- The use of a natural AP could reduce interference in the analysis. Synthetic protectants might introduce additional compounds that could interfere with the detection or identification of the target analyte. 3- Plant-derived materials might have components that help in preserving sample integrity

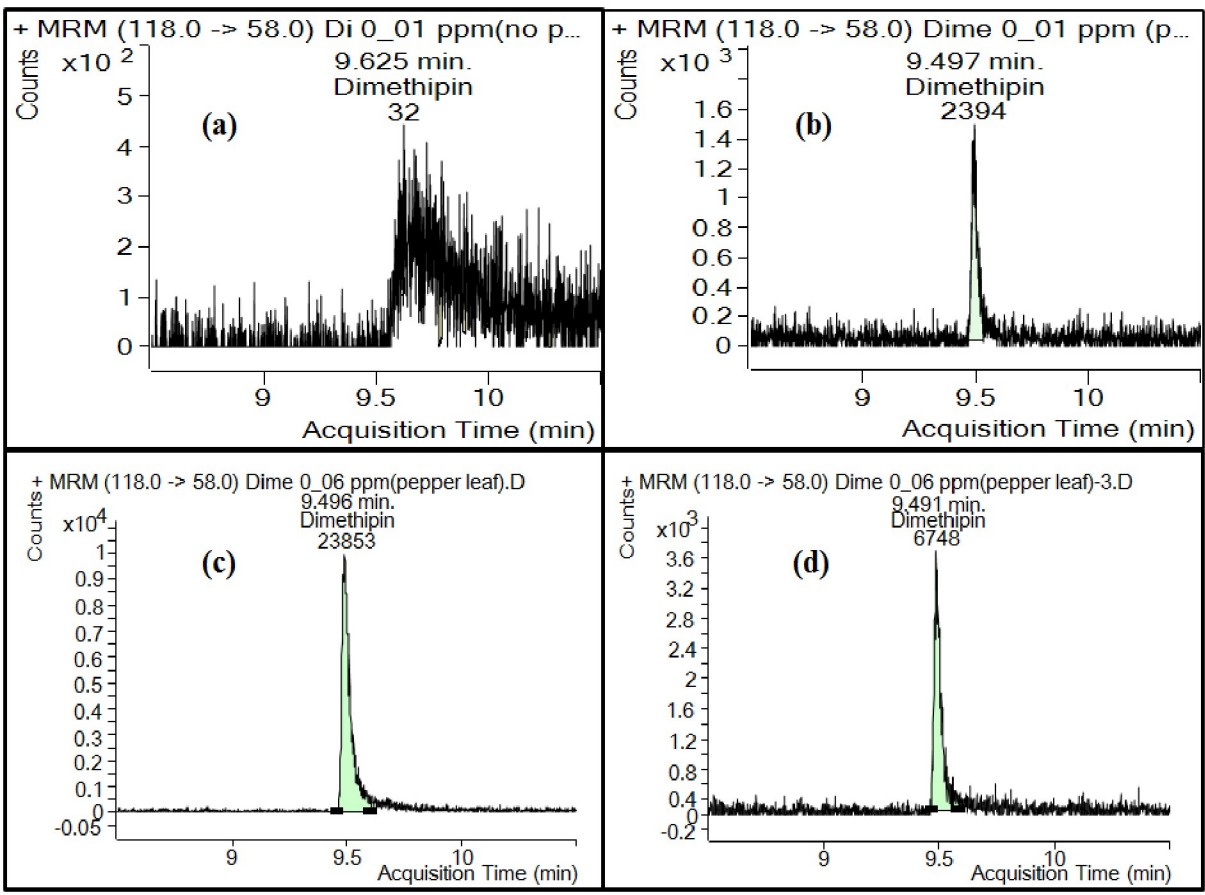

**Fig 1.** Effect of pepper leaf matrix on dimethipin signal at GC–MS/MS; (a) dimethipin 0.01 ppm without pepper leaf (b) dimethipin 0.01 ppm with pepper leaf matrix (c) dimethipin 0.06 ppm with pepper leaf matrix (d) signal suppression of dimethipin 0.06 ppm with pepper leaf matrix after three hours of continuous batch running.

during the analysis. This is especially important when dealing with complex matrices, such as food products. 4- Pepper leaves, being a natural product, may be more cost-effective than synthetic alternatives. 5- The use of a natural product aligns with eco-friendly practices, which is increasingly important in analytical chemistry. 6- The specific chemical composition of pepper leaves might offer unique properties that enhance the stability of the analyte during analysis.

## Method performance

Fig 3 displays representative GC–MS/MS chromatograms of matrix-matched standards, blank samples, and spiked blank samples of dimethipin with various analytes. Specificity was ensured by examining the presence of noise or interference at the retention times of the target analytes in the blank sample. An absence of signal or interference below 30% of the limit of quantification (LOQ) at the retention times of the target compounds indicated the method's specificity and minimal interference.

The matrix-matched calibration exhibited excellent linearity, with $R^2$ values ranging from 0.9988 to 0.9997 (Table 1). This indicates a strong linear relationship between the concentration of dimethipin and the corresponding response in the analytical method. The LOQ for dimethipin was determined to be 0.005 mg/kg (Table 1). The low LOQ values indicate the

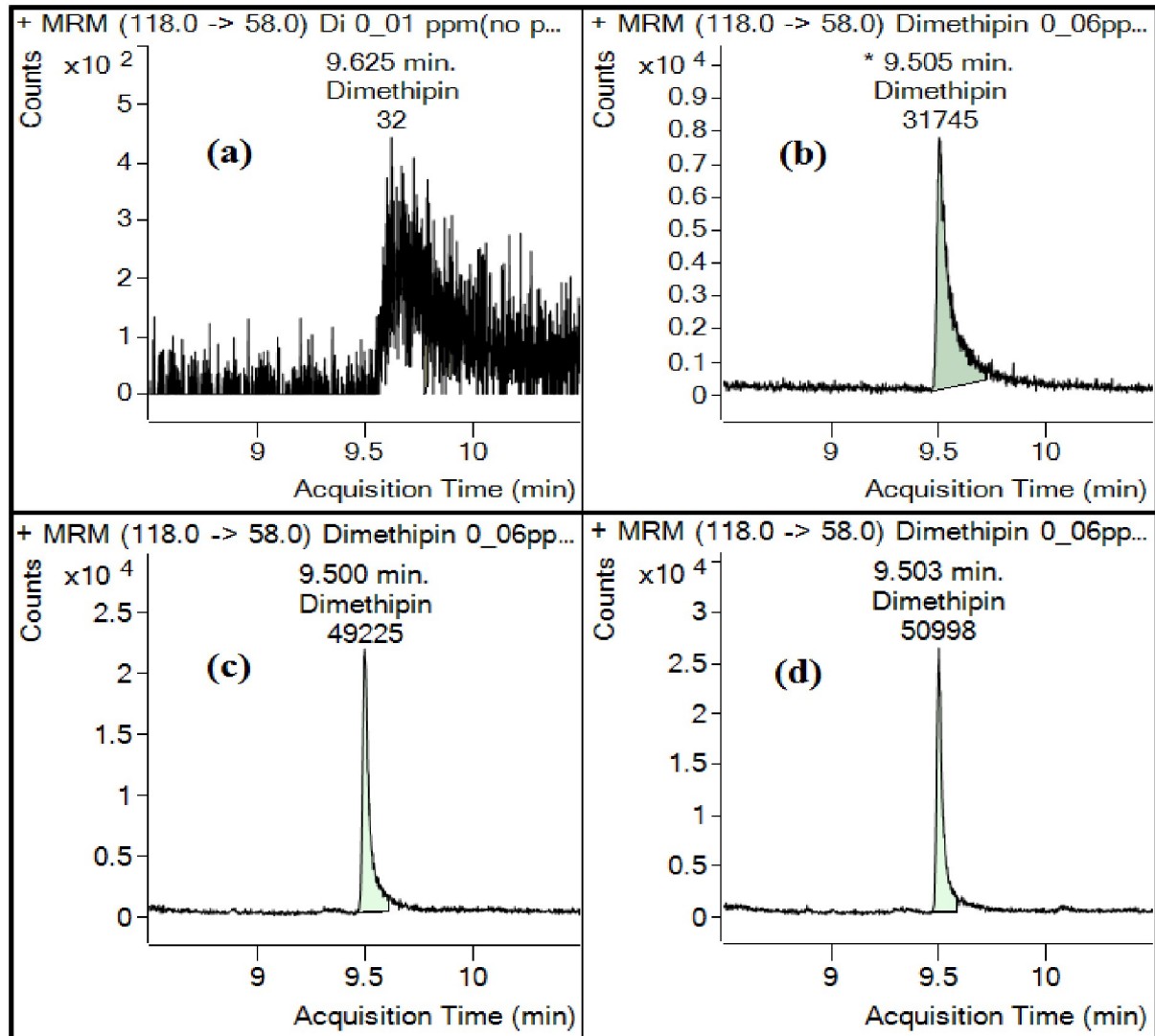

**Fig 2.** Optimization of analyte protectant (AP) for GC–MS/MS signal enhancement of dimethipin after priming with pepper leaf matrix (a) no priming (b) 10% AP after priming (c) 20% AP after priming (d) 30% AP after priming.

sensitivity and reliability of the developed method, allowing for the accurate detection and quantification of the herbicide even at very low concentrations within complex matrices. The practical benefits may enable the early detection of the herbicide, which is crucial for preventing and managing potential contamination in various complex matrices. Furthermore, it may indicate that our method complies with the regulations set by regulatory bodies (stringent limits), providing reliable data for regulatory purposes. It could also be essential for quality control and accurate assessment of the risk associated with herbicide exposure.

Table 1 presents the recovery rates and RSD values for dimethipin. All analytes exhibited recovery rates ranging from 88.8 to 110.0%, which fell within the acceptable range of 70–120% according to the SANTE guidance document on analytical quality control and method validation procedures for pesticide residue analysis in food and feed 2015 [37]. The observed precision values ranged from 1.4% to 12.0%, which were considered acceptable, as they fell within

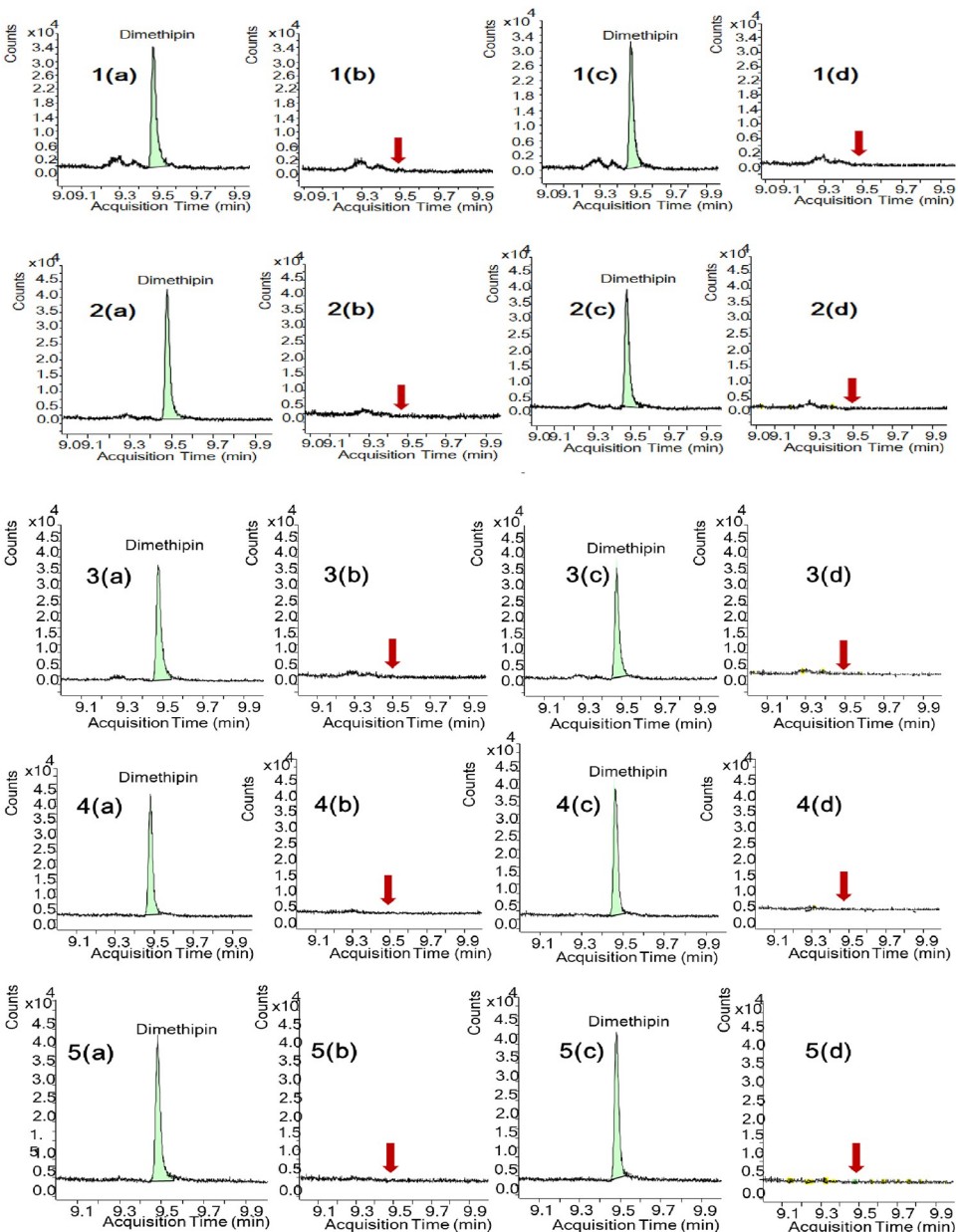

**Fig 3.** GC–MS/MS chromatograms of dimethipin in chicken (1), pork (2), beef (3), egg (4), and milk (5); a: standard in matrix (0.005 mg/kg), b: blank, c: recovery equivalent to 0.05 mg/kg, and d: market sample.

the coefficient of variation (20%). In this context, Rahman et al. [40] determined methiocarb and its metabolites in livestock products using liquid chromatography-tandem mass spectrometry. The method was validated at three fortification levels, yielding recovery rates ranging from 76.4% to 118.0% and relative standard deviations below 10.0%. These data suggest that the developed analytical method is effective for analyzing dimethipin in various matrices.

Across all matrices, dimethipin showed minimal matrix enhancement, ranging from +3.0% to +8.7% (Table 1). The matrix effects observed are within an acceptable range, indicating that the method accounts for the matrix composition to achieve accurate and reliable results.

**Table 1. Linearity, matrix effect, and recovery of dimethipin in animal-based food products.**

| Compound | Sample | Linearity ($R^2$) | Matrix effect (0.5 g/mL) | Recovery (n = 5) (Average ± RSD%) | | |
|---|---|---|---|---|---|---|
| | | | | LOQ | 2×LOQ | 10×LOQ |
| Dimethipin | Chicken | 0.9992 | +5.0±11.9 | 95.0 (10.7) | 91.5 (5.0) | 95.9 (1.8) |
| | Pork | 0.9988 | +8.7±13.5 | 97.5 (12.0) | 96.1 (5.0) | 96.3 (2.3) |
| | Beef | 0.9989 | +6.2±7.6 | 106.4 (8.1) | 105.1 (5.1) | 92.8 (8.8) |
| | Egg | 0.9997 | +3.0 ±4.3 | 110.0 (6.2) | 97.2 (4.7) | 101.6 (4.0) |
| | Milk | 0.9997 | +6.6±3.4 | 103.0 (6.7) | 97.0 (3.3) | 88.8 (1.4) |

Limit of quantification (LOQ): 0.005 mg/kg.

## Method application

After establishing the methodology, the developed method was effectively applied to analyze five different types of animal products (chicken, pork, beef, egg, and milk) acquired in 2019 and obtained from various markets across the Republic of Korea. The meat and egg samples were collected from nonorganic livestock markets in ten different locations, including Daejeon, Gyeongsan, Jeonbuk, Jeongup, Muan, Cheonan, Hongseong, Jangheung, Gyeonggi-do Gwangju, and Namwon. The milk samples represented ten major brands of milk produced in the Republic of Korea: Seul Milk, Maeil Milk, Home Plus, Pasteur Milk, Got Taste & Fresh Tech, Ildong, Purmil, Seoul Baby Milk, E-mart, and Sangha Farm. Importantly, none of the tested samples showed any positive results for the presence of dimethipin (Fig 3).

## Conclusion

In conclusion, a novel analytical method was developed using GC–MS/MS to detect dimethipin in animal-based food products. The technique employed priming with a pepper leaf matrix and an analyte protectant (AP) to enhance sensitivity and prevent thermal degradation. The method demonstrated a low limit of quantification (LOQ) of 0.005 mg/kg in all tested matrices. By synergistically integrating priming and analyte protectant strategies, we anticipate significant advancements in the sensitive detection of dimethipin residues in animal-based food products. This method offers a reliable and routine monitoring tool for detecting and quantifying dimethipin residues, providing valuable information for food safety and regulatory purposes.

## Acknowledgments

The publication of this article was funded by the Open Access Fund of Leibniz Universität Hannover.

## Author Contributions

**Conceptualization:** Fatih Oz.

**Formal analysis:** Jae-Han Shim, Tuba Esatbeyoglu, A. M. Abd El-Aty.

**Funding acquisition:** Tuba Esatbeyoglu, A. M. Abd El-Aty.

**Methodology:** Md. Musfiqur Rahman.

**Validation:** Jae-Han Shim, A. M. Abd El-Aty.

**Writing – review & editing:** Jae-Han Shim, Md. Musfiqur Rahman, Tuba Esatbeyoglu, Fatih Oz, A. M. Abd El-Aty.

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
