## [Decision Letter · Decision Letter 0]

10 Oct 2023

PONE-D-23-28489Development of an Enhanced Analytical Method Utilizing Pepper Matrix as an Analyte Protectant for Sensitive GC‒MSMS Detection of Dimethipin in Animal-Based Food ProductsPLOS ONE

Dear Dr. Esatbeyoglu,

Thank you for submitting your manuscript to PLOS ONE. After careful consideration, we feel that it has merit but does not fully meet PLOS ONE’s publication criteria as it currently stands. Therefore, we invite you to submit a revised version of the manuscript that addresses the points raised during the review process.

We look forward to receiving your revised manuscript.

Kind regards,

Benito Soto-Blanco, DVM, MSc, PhD

Academic Editor

PLOS ONE

- https://doi.org/10.1016/j.jchromb.2017.06.025

In your revision ensure you cite all your sources (including your own works), and quote or rephrase any duplicated text outside the methods section. Further consideration is dependent on these concerns being addressed.

Reviewers' comments:

Reviewer's Responses to Questions

**Comments to the Author**

1. Is the manuscript technically sound, and do the data support the conclusions?

Reviewer #1: Yes

Reviewer #2: Yes

2. Has the statistical analysis been performed appropriately and rigorously? 

Reviewer #1: No

Reviewer #2: Yes

3. Have the authors made all data underlying the findings in their manuscript fully available?

Reviewer #1: No

Reviewer #2: Yes

4. Is the manuscript presented in an intelligible fashion and written in standard English?

Reviewer #1: Yes

Reviewer #2: Yes

5. Review Comments to the Author

Reviewer #1: The manuscript titled: “Development of an enhanced analytical methods utilizing pepper matrix as an analyte protectant for sensitive GC-MSMS detection of dimethipin in animal-based food products” is an interesting study with promising analytical applications. The text is composed in a fluent and accessible manner, making it suitable for a wide readership. However, the material and method section lacks some information and structure in the discussion section (major revisions). Please consider the comments and suggestions below to improve the manuscript:

General comment:

-the authors should explain the advantage of using the pepper leaf as an AP over other ones (e.g., 3-ethoxy-1,2-propanediol, D-sorbitol);

-Indicate how was the pepper leaf prepared and standardized over the different samples;

- I suggest including statistical analyses (e.g. F-test for the linearity results, and ANOVA to check the elution and differences between the QuEChERS methods);

- Discussion: please improve the connection between your results and previous works (e.g, lines 273-279, 292-295).

Specific comments:

Title: keep the vertical bar between the MSMS as MS/MS;

Keywords: use complementary words to the title; that will help to track your article in the search engines;

Lines 52-60: Since the study is focused on this specific herbicide, it would be interesting to include information related to its use not only in the USA but in other locations (Europe, South America, Asia, and Africa). It would increase the importance of this study;

Line 79: indicate the range of concentration units that it can be found;

Line 123: indicate between brackets the 2 articles. The same for the article in line 126;

Line 147: please confirm if the mg of dimethipin used is 10.10 mg. In case the authors used the 10.10 mg, please correct the stock ppm values;

Lines 149-150: indicate here if this is a matrix-matched calibration curve;

Lines 151-152: please indicate for how long were these solutions stored;

Line 153: in the case of the spike samples, explain how they were done;

Line 154: indicate if the matrices were fresh or freeze-dried;

Line 164: Please, indicate the final concentration factor for the samples;

Line 171: indicate which type of liner was used in the injector port and the type of septum;

Lines 185: Indicate how often were the liner, septum, and column changed (or cut in the case of the last one); also it would be important to explain how the authors control the quality of the injections without the presence of a deuterated compound;

Line 189: indicate in this section how the authors calculate the matrix effect;

Line 192: indicate if the concentrations are expressed in FW;

Line 205: in this section, the authors only mention the results in the solvent and egg matrix; please clarify if the optimizations were done only in those 2 matrices or for all of them;

Lines 210-212: This sentence is confusing; please reformulate it. Besides this information contradicts the information mentioned in lines 212-213;

Lines 229-257: If this step was performed before the elution step and the selection of the best QuEChERS method, please change the order of these sections;

Figure 3: indicate at which LOQ level were those tests performed; indicate also the value of the integrated areas;

Lines 272-273: to improve the discussion section, please consider discussing the acquired LOQ values in terms of practicability to measure the herbicide in complex matrix samples;

Lines 273-279/292-295: Please connect the information with the results of the article to improve the discussion section;

Lines 280-281: this sentence should be in the material e methods, in the method validation section; the same for 284-285;

Table 1: Please include the accuracy results too;

Line 302: indicate when were the samples bought; indicate here or in the method validation section which quality controls were used within samples and/or batch injections;

Line 314: please substitute GC/MS/MS with GC-MS/MS.

Reviewer #2: This manuscript falls under the PLOS ONE scope and presents findings of research title “Development of an Enhanced Analytical Method Utilizing Pepper Matrix as an Analyte Protectant for Sensitive GC‒MSMS Detection of Dimethipin in Animal-Based Food Products”. The manuscript consists of 24 pages, 3 figures and 1 table. The paper presents interesting results as well as an inquisitive and reliable interpretation of the research results. The topic original and relevant in the field of study. The Abstract provides the highlights of the key contents of the main text. The Introduction provides enough background information to justify the study. The Results are consistent with the declared methodology, presented clearly enough, supported by the figures and tables. Researchers devised and validated analytical method using GC-MS/MS detection of dimethipin, and concluded that this method effectively addresses the challenge of analyzing dimethipin and can be used as a routine monitoring tool for dimethipin across various matrices. The methodology adequately described and conclusion consistent with the evidence and arguments presented. The references are appropriate and relevant to the research. However, minor typographical and grammatical errors need addressing.

6. PLOS authors have the option to publish the peer review history of their article (what does this mean?). If published, this will include your full peer review and any attached files.

Reviewer #1: **Yes: **Catarina da Rocha Cruzeiro

Reviewer #2: No

---

## [Author Response · Author response to Decision Letter 0]

3 Nov 2023

Thank you. This has been done

- https://doi.org/10.1016/j.jchromb.2017.06.025

Thank you. A slight overlap might occur due to similarities in authors' names, affiliations, chemicals and reagents, and common terminology, which is unavoidable.

In your revision ensure you cite all your sources (including your own works), and quote or rephrase any duplicated text outside the methods section. Further consideration is dependent on these concerns being addressed.

Done

“The funders had no role in the study design, data collection and analysis, decision to publish, or preparation of the manuscript.”

The authors received no specific funding for this work. We received only funding for the open access publication and stated within the manuscript as “The publication of this article was funded by the Open Access Fund of Leibniz Universität Hannover.”

"Upon resubmitting your revised manuscript, please upload your study’s minimal underlying data set as either Supporting Information files or to a stable, public repository and include the relevant URLs, DOIs, or accession numbers within your revised cover letter. For a list of acceptable repositories, please see http://journals.plos.org/plosone/s/data-availability#loc-recommended-repositories. Any potentially identifying patient information must be fully anonymized.

Thank you.

Reviewers' comments:

Reviewer #1: The manuscript titled: “Development of an enhanced analytical methods utilizing pepper matrix as an analyte protectant for sensitive GC‒MSMS detection of dimethipin in animal-based food products” is an interesting study with promising analytical applications. The text is composed in a fluent and accessible manner, making it suitable for a wide readership. However, the material and method section lacks some information and structure in the discussion section (major revisions). Please consider the comments and suggestions below to improve the manuscript:

We sincerely appreciate your thoughtful evaluation and constructive criticism of our manuscript titled "Development of an Enhanced Analytical Method Utilizing Pepper Matrix as an Analyte Protectant for Sensitive GC‒MS/MS Detection of Dimethipin in Animal-Based Food Products." Your insights have significantly contributed to enhancing the overall quality of our manuscript.

We have diligently addressed each of the comments and suggestions. Our aim was to offer a comprehensive and satisfactory response to your concerns, ensuring that the revised material and method section, as well as the discussion section, meet the standards expected for publication. We trust that the revisions made align with your expectations and contribute to the overall improvement of the manuscript.

Once again, we appreciate your meticulous review, and we remain open to any additional suggestions or feedback you may have. Your dedication to ensuring the rigor and clarity of scientific publications is highly valued.

General comment:

-the authors should explain the advantage of using the pepper leaf as an AP over other ones (e.g., 3-ethoxy-1,2-propanediol, D-sorbitol);

Thank you. This paragraph has been added to the text.

The choice of pepper leaf as an analyte protectant over alternatives, such as 3-ethoxy-1,2-propanediol or D-sorbitol, may offer several advantages:

1- Pepper leaf is a natural, plant-derived material, which might make it more compatible with certain analytes or matrices compared to synthetic compounds such as 3-ethoxy-1,2-propanediol.

2- The use of a natural analyte protectant could reduce interference in the analysis. Synthetic protectants might introduce additional compounds that could interfere with the detection or identification of the target analyte.

3- Plant-derived materials might have components that help in preserving sample integrity during the analysis. This is especially important when dealing with complex matrices, such as food products.

4- Pepper leaves, being a natural product, may be more cost-effective than synthetic alternatives.

5- The use of a natural product aligns with eco-friendly practices, which is increasingly important in analytical chemistry.

6- The specific chemical composition of pepper leaves might offer unique properties that enhance the stability of the analyte during analysis.

-Indicate how was the pepper leaf prepared and standardized over the different samples

This section has been conducted in accordance with our prior studies (Rahman et al., 2012; 2013; 2014)

- I suggest including statistical analyses (e.g. F test for the linearity results, and ANOVA to check the elution and differences between the QuEChERS methods);

Thank you for your suggestion. While we appreciate your emphasis on statistical analyses, we believe that it is not necessary to include them in this context. Our decision is based on the specific characteristics of our study and the nature of the data obtained.

- Discussion: please improve the connection between your results and previous works (e.g, lines 273-279, 292-295).

Thank you. This has been done

Specific comments:

Title: Keep the vertical bar between the MSMS as MS/MS;

This has been done

Keywords: use complementary words to the title; that will help to track your article in the search engines;

This has been amended as suggested.

Lines 52-60: Since the study is focused on this specific herbicide, it would be interesting to include information related to its use not only in the USA but in other locations (Europe, South America, Asia, and Africa). It would increase the importance of this study;

Thank you. Since the mid-1990s, the Americas have consistently held the position of the largest consumer of pesticides compared to other regions, surpassing Asia, Europe, Africa, and Oceania. As a result, the majority of reports tend to concentrate on the states rather than other regions.

Ref. Pesticides use and trade 1990–2021 FAOSTAT Analytical Brief 70

https://www.fao.org/3/cc6958en/cc6958en.pdf

Line 79: indicates the range of concentration units that can be found;

Thank you. There are currently no data available on the residual levels in products derived from livestock.

https://www3.epa.gov/pesticides/chem_search/reg_actions/reregistration/fs_PC-118901_1-Sep-05.pdf

Line 123: indicate between brackets the 2 articles. The same for the article in line 126;

Thank you. This has been added.

Line 147: please confirm if the mg of dimethipin used is 10.10 mg. In case the authors used the 10.10 mg, please correct the stock ppm values;

Yes, that is accurate. It is necessary to measure 10.10 mg from the dimethipin standard with a purity of 99%.

Lines 149-150: indicate here if this is a matrix-matched calibration curve;

Thank you. It is solvent calibration

Lines 151-152: please indicate for how long were these solutions stored;

Thank you. It is stored for a month.

Line 153: in the case of the spike samples, explain how they were done;

Thank you. This has been justified as follows: A total of 5 g of freeze-dried (chicken, pork, and beef) or fresh (egg and milk) samples was placed in a 50 mL Teflon centrifuge tube and spiked with 1 mL of 0.1 mg/kg (1 × limit of quantification [LOQ] level), 0.2 mg/kg (2 × LOQ level), or 1.0 mg/kg (10 × LOQ level).

Line 154: indicate if the matrices were fresh or freeze-dried

Thank you. The samples for chicken, pork, and beef were subjected to freeze-drying, while fresh samples were used for egg and milk.

Line 164: Please, indicate the final concentration factor for the samples

Thank you. The final concentration factor for the sample is 0.93. The concentration of AP was 30% (0.25 g/mL).

Line 171: indicate which type of liner was used in the injector port and the type of septum

Thank you. This information was added to the text. “The liner that was used in the injector port was Ultra inert liner, splitless single taper liner with glass wool (900 µL, 4 mm id (Part No. 5190-3163), whereas the septum was Inlet Septa: Agilent, Bleed/Temp optimized (BTO) nonstick 11 mm, 50/pk (Part No. 5183-4757)”.

Lines 185: Indicate how often were the liner, septum, and column changed (or cut in the case of the last one); also it would be important to explain how the authors control the quality of the injections without the presence of a deuterated compound;

Thank you. The frequency of change is twice a month for both the liner and septum, while for the column, it is once a month (1st cutting).

Controlling the quality of injections in GC‒MS/MS without the presence of a deuterated compound involves several strategies:

1. Peak Shape and Resolution: Well-defined, symmetrical peaks with good resolution indicate a high-quality injection.

2. Baseline Stability: A stable baseline suggests proper injection and chromatographic conditions.

3. Retention Time Stability: Consistent retention times across injections indicate reproducibility.

4. Blank Runs: Injecting blanks (solvent only) to ensure that there is no contamination or interference. The baseline should remain stable.

5. Standard Solutions: Standard solutions were injected periodically to verify the instrument's sensitivity and linearity.

6. Matrix Effects: Evaluating the matrix effects by analyzing matrix-matched standards. This helps assess the impact of the sample matrix on the analytical signal.

7. Calibration standards: Calibration standards were injected regularly to ensure that the instrument response was consistent with the calibration curve.

8. Peak Identification: Confirming peak identity by comparing mass spectra and retention times with reference standards or library spectra.

Line 189: indicate in this section how the authors calculate the matrix effect;

Thank you. This has been added.

Line 192: indicate if the concentrations are expressed in FW;

Yes, it is expressed as fresh weight.

Line 205: in this section, the authors only mention the results in the solvent and egg matrix; please clarify if the optimizations were done only in those 2 matrices or for all of them;

Thank you. Yes, the process is conducted using solvent and egg matrix exclusively, serving as representatives for other matrices.

Lines 210-212: This sentence is confusing; please reformulate it. In addition, this information contradicts the information mentioned in lines 212-213;

Thank you. The highest was replaced with “satisfactory”.

Lines 229-257: If this step was performed before the elution step and the selection of the best QuEChERS method, please change the order of these sections;

Thank you. No, it was conducted after elution

Figure 3: indicate at which LOQ level were those tests performed; indicate also the value of the integrated areas;

Thank you. The caption has been revised to “Figure 3. GC‒MS/MS chromatograms of dimethipin in chicken (1), pork (2), beef (3), egg (4), and milk (5); a: standard in matrix (0.005 mg/kg), b: blank, c: recovery equivalent to 0.05 mg/kg, and d: market sample.

Lines 272-273: to improve the discussion section, please consider discussing the acquired LOQ values in terms of practicability to measure the herbicide in complex matrix samples;

Thank you. This has been amended as suggested.

Lines 273-279/292-295: Please connect the information with the results of the article to improve the discussion section;

Thank you. This has been done

Lines 280-281: this sentence should be in the material e methods, in the method validation section; the same for 284-285;

Thank you. This has been amended as suggested.

Table 1: Please include the accuracy results too;

Thank you. The accuracy results are indicated by the recovery rates.

Line 302: indicate when were the samples bought; indicate here or in the method validation section which quality controls were used within samples and/or batch injections;

Thank you. The samples were acquired in 2019.

The QCs can be implemented through calibration standards, including blanks in each batch to monitor and correct for any contamination or background signals, and matrix-matched standards.

Line 314: please substitute GC/MS/MS with GC‒MS/MS.

Thank you. This has been amended.

Response to Reviewer #2 comments:

This manuscript falls under the PLOS ONE scope and presents findings of research title “Development of an Enhanced Analytical Method Utilizing Pepper Matrix as an Analyte Protectant for Sensitive GC‒MSMS Detection of Dimethipin in Animal-Based Food Products”. The manuscript consists of 24 pages, 3 figures and 1 table. The paper presents interesting results as well as an inquisitive and reliable interpretation of the research results. The topic original and relevant in the field of study. The Abstract provides the highlights of the key contents of the main text. The Introduction provides enough background information to justify the study. The Results are consistent with the declared methodology, presented clearly enough, supported by the figures and tables. Researchers devised and validated an analytical method using GC‒MS/MS detection of dimethipin and concluded that this method effectively addresses the challenge of analyzing dimethipin and can be used as a routine monitoring tool for dimethipin across various matrices. The methodology adequately described and conclusion consistent with the evidence and arguments presented. The references are appropriate and relevant to the research. However, minor typographical and grammatical errors need addressing.

Thank you. We have checked the text once again for grammar and syntax errors.

---

## [Decision Letter · Decision Letter 1]

13 Nov 2023

PONE-D-23-28489R1Development of an Enhanced Analytical Method Utilizing Pepper Matrix as an Analyte Protectant for Sensitive GC‒MS/MS Detection of Dimethipin in Animal-Based Food ProductsPLOS ONE

Dear Dr. Esatbeyoglu,

Thank you for submitting your manuscript to PLOS ONE. After careful consideration, we feel that it has merit but does not fully meet PLOS ONE’s publication criteria as it currently stands. Therefore, we invite you to submit a revised version of the manuscript that addresses the points raised during the review process.

We look forward to receiving your revised manuscript.

Kind regards,

Benito Soto-Blanco, DVM, MSc, PhD

Academic Editor

PLOS ONE

Journal Requirements:

Reviewers' comments:

Reviewer's Responses to Questions

**Comments to the Author**

1. If the authors have adequately addressed your comments raised in a previous round of review and you feel that this manuscript is now acceptable for publication, you may indicate that here to bypass the “Comments to the Author” section, enter your conflict of interest statement in the “Confidential to Editor” section, and submit your "Accept" recommendation.

Reviewer #1: (No Response)

Reviewer #2: All comments have been addressed

2. Is the manuscript technically sound, and do the data support the conclusions?

Reviewer #1: Yes

Reviewer #2: (No Response)

3. Has the statistical analysis been performed appropriately and rigorously? 

Reviewer #1: No

Reviewer #2: (No Response)

4. Have the authors made all data underlying the findings in their manuscript fully available?

Reviewer #1: No

Reviewer #2: (No Response)

5. Is the manuscript presented in an intelligible fashion and written in standard English?

Reviewer #1: Yes

Reviewer #2: (No Response)

6. Review Comments to the Author

Reviewer #1: The authors have responded to all queries raised by the reviewer, enhancing the overall quality of the manuscript. Prior to acceptance, kindly incorporate some of the provided responses (indicated as R below) to elucidate these aspects. This clarification aims not only to address the concerns raised by the reviewer but also to provide enhanced clarity for all readers.

Original:Lines 52-60: Since the study is focused on this specific herbicide, it would interesting to include information related to its use not only in the USA but in other locations (Europe, South America, Asia, and Africa). It would increase the importance of this study;

R: Thank you. Since the mid-1990s, the Americas have consistently held the position of the largest consumer of pesticides compared to other regions, surpassing Asia, Europe, Africa, and Oceania. As a result, the majority of reports tend to concentrate on the states rather than other regions. Ref. Pesticides use and trade 1990–2021 FAOSTAT Analytical Brief 70 https://www.fao.org/3/cc6958en/cc6958en.pdf

-Please add that information to the manuscript

Original:Line 164: Please, indicate the final concentration factor for the samples

R: Thank you. The final concentration factor for the sample is 0.93. The concentration of AP was 30% (0.25 g/mL).

-Please indicate that information in the manuscript

Original:Lines 185: Indicate how often were the liner, septum, and column changed (or cut in the case of the last one); also it would be important to explain how the authors control the quality of the injections without the presence of a deuterated compound;

R: Thank you. The frequency of change is twice a month for both the liner and septum, while for the column, it is once a month (1st cutting).

-Please indicate that information in the manuscript

R: Controlling the quality of injections in GC‒MS/MS without the presence of a deuterated compound involves several strategies:…..

-Use that information in the discussion to support the quality of your analyses

Original:Line 192: indicate if the concentrations are expressed in FW;

R: Yes, it is expressed as fresh weight

-Please add that information in the manuscript.

Original:Line 302: indicate when were the samples bought; indicate here or in the method validation section which quality controls were used within samples and/or batch injections;

R: 1)Thank you. The samples were acquired in 2019.

R: 2)The QCs can be implemented through calibration standards, including blanks in each batch to monitor and correct for any contamination or background signals, and matrix matched standards.

-1) please indicate that information in the method application section

-2) please indicate that information in the material and methods section

Reviewer #2: (No Response)

7. PLOS authors have the option to publish the peer review history of their article (what does this mean?). If published, this will include your full peer review and any attached files.

Reviewer #1: **Yes: **Catarina Cruzeiro

Reviewer #2: **Yes: **Issa S. Al-Amri

---

## [Author Response · Author response to Decision Letter 1]

15 Nov 2023

Response to the journal and reviewer’s comments

Journal Requirements

Please review your reference list to ensure that it is complete and correct. If you have cited papers that have been retracted, please include the rationale for doing so in the manuscript text, or remove these references and replace them with relevant current references. Any changes to the reference list should be mentioned in the rebuttal letter that accompanies your revised manuscript. If you need to cite a retracted article, indicate the article’s retracted status in the References list and include a citation and full reference for the retraction notice.

Response: Thank you for your thorough review of our manuscript and the reference list. We have carefully examined all cited references, and we can confirm that none of them were taken from retracted articles. The reference list has been reviewed for completeness and accuracy, and we believe it accurately represents the current state of the literature related to our work.

Reviewer # 1 comments: 

Reviewer #1: The authors have responded to all queries raised by the reviewer, enhancing the overall quality of the manuscript. Prior to acceptance, kindly incorporate some of the provided responses (indicated as R below) to elucidate these aspects. This clarification aims not only to address the concerns raised by the reviewer but also to provide enhanced clarity for all readers.

Original:Lines 52-60: Since the study is focused on this specific herbicide, it would interesting to include information related to its use not only in the USA but in other locations (Europe, South America, Asia, and Africa). It would increase the importance of this study;

R: Thank you. Since the mid-1990s, the Americas have consistently held the position of the largest consumer of pesticides compared to other regions, surpassing Asia, Europe, Africa, and Oceania. As a result, the majority of reports tend to concentrate on the states rather than other regions. Ref. Pesticides use and trade 1990–2021 FAOSTAT Analytical Brief 70 https://www.fao.org/3/cc6958en/cc6958en.pdf

-Please add that information to the manuscript

Original:Line 164: Please, indicate the final concentration factor for the samples

R: Thank you. The final concentration factor for the sample is 0.93. The concentration of AP was 30% (0.25 g/mL).

-Please indicate that information in the manuscript

Original:Lines 185: Indicate how often were the liner, septum, and column changed (or cut in the case of the last one); also it would be important to explain how the authors control the quality of the injections without the presence of a deuterated compound;

R: Thank you. The frequency of change is twice a month for both the liner and septum, while for the column, it is once a month (1st cutting).

-Please indicate that information in the manuscript

R: Controlling the quality of injections in GC‒MS/MS without the presence of a deuterated compound involves several strategies:…..

-Use that information in the discussion to support the quality of your analyses

Original:Line 192: indicate if the concentrations are expressed in FW;

R: Yes, it is expressed as fresh weight

-Please add that information in the manuscript.

Original:Line 302: indicate when were the samples bought; indicate here or in the method validation section which quality controls were used within samples and/or batch injections;

R: 1)Thank you. The samples were acquired in 2019.

R: 2)The QCs can be implemented through calibration standards, including blanks in each batch to monitor and correct for any contamination or background signals, and matrix matched standards.

-1) please indicate that information in the method application section

-2) please indicate that information in the material and methods section

Response: Thank you to the diligent reviewer for the feedback. We have carefully reviewed the comments and suggestions and have incorporated the necessary additions and clarifications as indicated by "R" in the revised manuscript. We believe that these changes enhance the overall clarity of the manuscript for all readers, as denoted by the esteemed reviewer.

---

## [Decision Letter · Decision Letter 2]

4 Dec 2023

Development of an Enhanced Analytical Method Utilizing Pepper Matrix as an Analyte Protectant for Sensitive GC‒MS/MS Detection of Dimethipin in Animal-Based Food Products

PONE-D-23-28489R2

Dear Dr. Esatbeyoglu,

We’re pleased to inform you that your manuscript has been judged scientifically suitable for publication and will be formally accepted for publication once it meets all outstanding technical requirements.

Kind regards,

Benito Soto-Blanco, DVM, MSc, PhD

Academic Editor

PLOS ONE

Reviewers' comments:

Reviewer's Responses to Questions

**Comments to the Author**

1. If the authors have adequately addressed your comments raised in a previous round of review and you feel that this manuscript is now acceptable for publication, you may indicate that here to bypass the “Comments to the Author” section, enter your conflict of interest statement in the “Confidential to Editor” section, and submit your "Accept" recommendation.

Reviewer #1: All comments have been addressed

Reviewer #2: All comments have been addressed

2. Is the manuscript technically sound, and do the data support the conclusions?

Reviewer #1: Yes

Reviewer #2: (No Response)

3. Has the statistical analysis been performed appropriately and rigorously? 

Reviewer #1: No

Reviewer #2: (No Response)

4. Have the authors made all data underlying the findings in their manuscript fully available?

Reviewer #1: No

Reviewer #2: (No Response)

5. Is the manuscript presented in an intelligible fashion and written in standard English?

Reviewer #1: Yes

Reviewer #2: (No Response)

6. Review Comments to the Author

Reviewer #1: (No Response)

Reviewer #2: (No Response)

7. PLOS authors have the option to publish the peer review history of their article (what does this mean?). If published, this will include your full peer review and any attached files.

Reviewer #1: **Yes: **Catarina da Rocha Cruzeiro

Reviewer #2: **Yes: **Issa S. Al-Amri

---

## [Editor Report · Acceptance letter]

7 Dec 2023

PONE-D-23-28489R2 

Development of an Enhanced Analytical Method Utilizing Pepper Matrix as an Analyte Protectant for Sensitive GC‒MS/MS Detection of Dimethipin in Animal-Based Food Products 

Dear Dr. Esatbeyoglu:

I'm pleased to inform you that your manuscript has been deemed suitable for publication in PLOS ONE. Congratulations! Your manuscript is now with our production department. 

Kind regards, 

on behalf of

Dr. Benito Soto-Blanco 

Academic Editor

PLOS ONE